# AI and the Everything in the Whole Wide World Benchmark

**Inioluwa Deborah Raji**
Mozilla Foundation, UC Berkeley
rajiinio@berkeley.edu

**Emily M. Bender**
Department of Linguistics
University of Washington

**Amandalynne Paullada**
Department of Linguistics
University of Washington

**Emily Denton**
Google Research

**Alex Hanna**
Google Research

## Abstract

There is a tendency across different subfields in AI to valorize a small collection of influential benchmarks. These benchmarks operate as stand-ins for a range of anointed common problems that are frequently framed as foundational milestones on the path towards flexible and generalizable AI systems. State-of-the-art performance on these benchmarks is widely understood as indicative of progress towards these long-term goals. In this position paper, we explore the limits of such benchmarks in order to reveal the construct validity issues in their framing as the functionally "general" broad measures of progress they are set up to be.

## 1 Introduction

In the 1974 Sesame Street children's storybook *Grover and the Everything in the Whole Wide World Museum* [Stiles and Wilcox, 1974], the Muppet monster Grover visits a museum claiming to showcase "everything in the whole wide world". Example objects representing certain categories fill each room. Several categories are arbitrary and subjective, including showrooms for "Things You Find On a Wall" and "The Things that Can Tickle You Room". Some are oddly specific, such as "The Carrot Room", while others unhelpfully vague like "The Tall Hall". When he thinks that he has seen all that is there, Grover comes to a door that is labeled "Everything Else". He opens the door, only to find himself in the outside world.

As a children's story, Grover's described situation is meant to be absurd. However, in this paper, we discuss how a similar faulty logic is inherent to recent trends in artificial intelligence (AI) — and specifically machine learning (ML) — evaluation, where many popular benchmarks rely on the same false assumptions inherent to the ridiculous "Everything in the Whole Wide World Museum" that Grover visits. In particular, we argue that benchmarks presented as measurements of progress towards general ability within vague tasks such as "visual understanding" or "language understanding" are as ineffective as the finite museum is at representing "everything in the whole wide world," and for similar reasons — being inherently specific, finite and contextual.

Benchmarks like GLUE [Wang et al., 2019a] or ImageNet [Deng et al., 2009] are often elevated to become definitions of the essential common tasks to validate the performance of any given model. As a result, often the claims that are justified through these benchmark datasets extend far beyond the tasks they are initially designed for, and reach beyond even the initial ambitions for development. Despite a presentation and acceptance as markers of progress towards general-purpose capabilities, there are clear limitations of these benchmarks. In fact, the reality of their development, use and adoption indicates a *construct validity* issue, where the involved benchmarks — due to their instantiation in particular data, metrics and practice — cannot possibly capture anything representative of the claims to general applicability being made about them. In this paper, we illuminate the over-reliance on the

35th Conference on Neural Information Processing Systems (NeurIPS 2021) Track on Datasets and Benchmarks.

historical Common Task Framework (CTF) for machine learning as it inappropriately evolved into what we understand today to be these benchmarks claiming to assess general capabilities. We do not deny the utility of such benchmarks, but rather hope to point to the risks inherent in their framing. Our goal is to dig into the details of what it means for us as a community to continue propping up certain benchmarks with the rhetoric of "general" performance.

## 2 Background

### 2.1 Striving for Generality

The notion of a "general-purpose" AI can be traced, in part, back to the early days of the field. In the late 1950s, Newell et al. [1959] proposed the "General Problem Solver" in an attempt to create a system that could solve a range of problems, as well as develop a general theory of problem solving. While this project ultimately failed, it helped establish general intelligence as a goal within the field of AI. As Adam [1998] notes, the failure of the project was at the time attributed primarily to implementation issues, rather than the perception of any flaw in the formulation of a disembodied and general-purpose problem solver as a meaningful goal.

Researchers now often use the term "generality" to refer to the development of AI systems that are flexible in nature, demonstrating competence on a wide range of tasks and in a wide range of settings, hoping to mirror the adaptive cognitive abilities humans are perceived to possess [Shevlin et al., 2019]. While discussions of "generality" might evoke the goal of artificial general intelligence [Voss, 2007, Pennachin and Goertzel, 2007], we set that aside in this paper and focus instead on the more immediate (and logically prior) issue of claims of generality for a specific type of cognitive skill or capability (e.g., vision or language), supposedly independent of context or application domains.

Within specific sub-fields and more pragmatic settings, researchers will often strive for the development of general-purpose systems that capture a breadth of functionality or knowledge. The development of general-purpose feature embeddings has also become a central research focus within machine learning communities. Here, the goal is the development of representations — through either unsupervised or supervised means — that can generalize (with minimal fine-tuning) to a wide range of other tasks they were not specifically developed for [e.g. Huh et al., 2016, Du et al., 2020, Bommasani et al., 2021].

### 2.2 A Brief History of Benchmarking Practice in AI

Some assess AI progress by asking scientific and philosophical questions on the notion of intelligence, thinking of AI as a method for modeling some general cognitive function. Others, particularly those in the machine learning field, evaluate performance based on the success of a model's utility within some practical applications. Although the former goals sparked the aspiration to measure general cognitive capabilities, the latter application-based objectives are what data benchmarking has been historically designed to communicate.

In this paper we describe a *benchmark* as a particular combination of a dataset or sets of datasets (at least test data, sometimes also training data), and a metric, conceptualized as representing one or more specific tasks or sets of abilities, picked up by a community of researchers as a shared framework for the comparison of methods. The *task* is a particular specification of a problem, as represented in the dataset.[1] A *metric* is a way to summarize system performance over some set or sets of tasks as a single number or score. The metric provides a means of counting success and failure at the level of individual system outputs and summarizing those counts over the full dataset. Models obtaining the most favourable scores on the metrics for a benchmark are considered to be "state-of-the-art" (SOTA) in terms of performance on the specified task. Here, we present a stylized history that's far from comprehensive, but attempts to provide the shape of the development of predecessors of current machine learning benchmarking practice.

Before benchmarks were employed for algorithm selection in machine learning, they were used for "computer selection" — i.e. the task of running a benchmark program on multiple computers to determine which machine was most suitable for purchase. In 1962, Auberbach Corporation's Standard EDP Reports were developed as a way to directly assess the performance of machines

---

[1]This corresponds to Schlangen's [2021] notion of extensional definition of tasks; see §4.1.

on specific prototype tasks, to avoid the tedious process of individually weighing the quality of the various features of computers offered by different vendors. By 1976, benchmarking was a fairly common practice in computing, with standard libraries of mock software programs created for the purpose of testing specific subroutines for core functional tasks (i.e. file management, word processing, etc.) being developed and used by the U.S. government as part of the assessment involved in their procurement process for computer systems [Lewis and Crews, 1985].

Several linguists [e.g. Liberman, 2010, Church, 2018] have traced the influence of Fred Jelinek at IBM and Charles Wayne at DARPA as crucial protagonists in the development of a common framework for the quantitative assessment of computational linguistic tasks, notably within speech recognition and machine translation. Beginning in the mid-1980s, Jelinek had been a driver in developing what he had called the Common Task Framework (CTF). The Common Task Framework, the precursor to modern data benchmarking in AI, was initially introduced in the 1980s as a response to Wendell Pierce's influential assertion of a field "being deceived by the glamour of (a would-be) theory, rather than actual performance" [Donoho, 2017, p.17]. The primary elements of the CTF include (a) A publicly available training dataset with a list of feature measurements and a class label for that observation; (b) A set of enrolled competitors whose common task is to infer a class prediction rule from the training data; and (c) A scoring referee, to which competitors can submit their prediction rule [Donoho, 2017, p. 572]. With DARPA's sponsorship and IBM's data, the CTF soon became widely adopted as a common mode of research in machine translation and speech recognition. Rather than experimental results that were updated "twice a year", researchers could evaluate their experiments every hour [Church, 2018], and, although major breakthroughs were still rare, claims to incremental improvement enabled by this "quantitative comparison of alternative algorithms on a fixed task", ensured that researchers could track steady progress regardless [Liberman, 2010]. Soon, similar NLP-related initiatives followed — such as Shared Task Evaluation Campaigns (STECs) [Belz and Kilgarriff, 2006], Message Understanding Conferences (MUCs) [Grishman and Sundheim, 1996], and Text REtrieval Conferences (TRECs) for information retrieval.[2] Later, STECs within NLP coalesced around machine translation (NIST-MT)[Przybocki et al., 2009] and word sense disambiguation (SENSEVAL)[Kilgarri, 1998].

Parallel developments also occurred in other fields. The Facial Recognition Technology (FERET) program for automated facial recognition, inaugurated in 1993 [Phillips et al., 2000], was one of the first to bring a CTF type evaluation approach to the computer vision. The National Institute of Standards (NIST) reports that, prior to the publication of their FERET database, "[o]nly a few of these algorithms reported results on images utilizing a common database let alone met the desirable goal of being evaluated on a standard testing protocol that included separate training and testing sets. As a consequence, there was no method to make informed comparisons among various algorithms" [Phillips et al., 2000, p. 2]. Meanwhile, modern reinforcement learning algorithms have traditionally anchored to a culture of benchmarking, often evaluated on gaming environments like Atari, Starcraft, Dota2, and Go. One of the first games to widely capture the imagination of AI researchers was chess which became a fixture of US-Soviet Cold War competition and a stand-in for national technological achievement. [Ensmenger, 2012, p. 6]. More generally, the UC Irvine Machine Learning Repository (UCI) [Dua and Graff, 2017] was created in 1987 as a response to many calls for machine learning to have a centralized location for data in machine learning [Radin, 2017] and has morphed into a repository for a wide array of tasks and data.

## 2.3 Construct Validity

If reproducibility and reliability are about the precision and thus the reliable repeatability of a finding, then challenges with validity for machine learning would be focused on how the field approaches the question of accuracy — namely, how closely our evaluations hit the mark in appropriately characterizing the actual anticipated behaviour of the system in the real world or progress on stated motivations and goals for the field. The accuracy of these evaluations is thus not just a question of reporting consistency but an actual design task involving reflection on what the objectives of an effective model involve, in addition to conscious decision making on how to best represent these desired outcomes in evaluation metrics, data and methodology. In particular, *construct validity* is an external validity issue related to how well an experimental setting relates to a research claim, or, in the context of machine learning, how well the benchmark dataset, and associated metrics of

---

[2]https://trec.nist.gov

evaluation, represents a task [O'Leary-Kelly and Vokurka, 1998]. It concerns how well designed — or rather, how well *constructed* — the experimental setting is in relation to the research claim. Validity issues have long been acknowledged broadly by the machine learning community as a particularly difficult task in benchmark design and development [Mitchell, 2021, Jacobs and Wallach, 2021, Malik, 2020] — however as Bowman and Dahl note, it remains a generally neglected issue as "[t]his criterion is difficult to fully formalize, and we know of no simple test that would allow one to determine if a benchmark presents a valid measure of model ability" [Bowman and Dahl, 2021, p. 3].

## 3  Attempting to Benchmark General Capabilities

We now examine a particular style of benchmark dataset — and the accompanying practices of use — that has gained in popularity in recent years. These benchmarks have been claimed to embed notions of generality, both in their presentation by the creators and the manner in which they are adopted and used by the machine learning community.

Even when the creators of such benchmarks do not explicitly purport to be establishing benchmarks for "general intelligence", community practices and overall hype have dramatized what it means for a model to perform well on these benchmarks. The inspiration for the setup of these benchmarks is often explicitly linked to the general-purpose capabilities and breadth of knowledge humans possess. In doing so, the ethos that surrounds these datasets often extends beyond a reasonable scope of interpretation and influence, especially given the limitations inherent in their construction.

In this paper, we argue that the aim of measuring general-purpose capabilities (i.e. goals such as general-purpose object recognition, general language understanding or domain-independent reasoning) cannot be adequately embodied in data-defined benchmarks. We observe that current trends inappropriately extend the Common Task Framework (CTF) paradigm to apply to the "task" of performance in the abstract, distinct from real world objectives or context. Historically, the CTF was developed precisely to introduce practically-oriented and tightly-scoped AI tasks — namely, automatic speech recognition (ASR) or machine translation (MT) — where the required validation is whether the benchmark accurately reflects the practical task being asked of the computer in its real-world context [Donoho, 2017]. This new wave of poorly defined "general" objectives completely subverts the intention of its introduction and actually enables its use to promote — rather than counteract — claims to "glamour and deceit" over substantial, meaningful progress.

### 3.1  Case Studies

We focus our analysis on two common benchmarks geared towards the assessment of general capabilities — ImageNet and GLUE, described below. The discussion is scoped to *dataset*-based benchmarks, and will not include much discussion of the role of gameplay demos in AI development.

#### 3.1.1  ImageNet

We characterize ImageNet as a "general" benchmark due to several related observations. First, at the time of development, the creators described ImageNet as representing "the most comprehensive and diverse coverage of the image world" [Deng et al., 2009, p.1] and, retrospectively, Li described the project as an "attempt to map the entire world of objects" [Gershgorn, 2017]. With general-purpose visual object recognition framed as the ability to recognize a sizable breadth of objects in a manner that rivals human capabilities [Liu et al., 2020], we observe ImageNet's sheer size — both in terms of number of categories and number of images per category — further informs its perception as representing a general formulation of visual object recognition. At the time of its creation, it offered 20 times the number of categories, and 100 times the number of total images as the most popular training and evaluation benchmarks at the time, such as Caltech 101/256 [Kamarudin et al., 2015, Griffin et al., 2007], MSRC [Shotton et al., 2006] and PASCAL VOC [Everingham et al., 2010]. Second, we observe community consensus that the task formalized in the ImageNet dataset represents a meaningful milestone towards longer-term goals of artificial visual intelligence. Indeed, Li has explicitly characterized large scale visual object recognition as the "north star" of computer vision — a scientific quest that would define and guide the field towards the ultimate goal of artificial visual intelligence [Li, 2019], with ImageNet operating as the canonical instantiation. In their papers, ImageNet authors are clear about the desire to situate the dataset as the definitive benchmark of the field. However, the hype surrounding ImageNet and the scope of the claims derived from it are

not localized the the field of computer vision. For example, increased performance on ImageNet is often explicitly referenced as an indication that the field is progressing towards general-purpose AI [Sutskever, 2018].

## 3.2   GLUE and SuperGLUE

The creators of GLUE (General Language Understanding Evaluation) [Wang et al., 2019a] and SuperGLUE [Wang et al., 2019b] present these resources as "evaluation framework[s] for research towards general-purpose language understanding technologies" [Wang et al., 2019b, p.1], noting that, unlike human language understanding, most computer natural language understanding (NLU) systems are task or domain-specific [Wang et al., 2019a]. When a human knows a language, they can use that knowledge across any task that involves that language. Thus, a benchmark that tests whether linguistic knowledge acquired through training on one task can be applied to other tasks, in principle, tests for a specific and potentially well-defined kind of generalizability.

Authors for GLUE and SuperGLUE describe both datasets as being "designed to provide a general-purpose evaluation of language understanding that covers a range of training data volumes, task genres, and task formulations" [Wang et al., 2019b, p.2]. In framing these benchmarks as general language understanding evaluation benchmarks made up of a diverse set of language understanding tasks, the authors suggest the notion that success on the particular tasks included demonstrates at least progress towards a full-scale solution to language understanding, and explicitly describe such a set up as being well positioned for "exhibiting the transfer-learning potential of approaches" [Wang et al., 2019b, p.2]. As a comprehensive setup for a wide range of tests in a popular language (English), this benchmark has been elevated to its current status as a general marker of progress for the NLP community, and specifically the accepted evaluation platform for "general-purpose sentence encoders" [Wang et al., 2019a].

# 4   Limits of Benchmarking General Capabilities

The imagined artifact of the "general" benchmark does not actually exist. Real data is designed, subjective and limited in ways that necessitate a different framing from that of any claim to general knowledge or general-purpose capabilities. In fact, presenting any single dataset in this way is ultimately dangerous and deceptive, resulting in misguidance on task design and focus, under-reporting of the many biases and subjective interpretations inherent in the data as well as enabling, through false presentations of performance, potential model misuse. In this section, we walk through the key arguments for how benchmarking is a limited approach to assess general model capabilities and, in particular, discuss the risk of making this claim to generality in the context of the *limited task design*, *de-contextualized data and performance reporting* as well as *inappropriate community use* common to such benchmarks in the ML context. For each of these limitations, we break down the shortcoming and discuss the cited evidence and reasoning for our observations.

## 4.1   Limited Task Design

There is nothing systematic or organized about the definition and arrangement of the rooms in the museum that Grover visits. He walks through arbitrary rooms of "Things That Make So Much Noise You Can't Hear Yourself Think", "The Small Hall", "The Carrot Room" and the like.

Similarly, the task formation for these benchmarks seems to happen independently of the intended and declared problem space. Schlangen [2021] defines a *task* as a mapping from an input space to an output space, defined both *intensionally* (via a description of the task) and *extensionally* (via a particular dataset, i.e. pairs of inputs and outputs matching the description). In machine learning, the tendency is towards the latter scenario, where the benchmark task is defined by a dataset, often collected with such a task in mind [Scheuerman et al., 2021]. If the so-called "general" benchmarks were legitimate tests of progress towards general artificial cognitive abilities, we would expect the tasks they embody to be chosen systematically, or with reference to specific theories of the cognitive abilities they model. Instead, what we observe looks more like samples of convenience: tasks and collections of tasks arbitrarily built out of what is easily available to the team developing these benchmarks, even if such constructions are theoretically unsound. Unlike with software benchmarking [Lewis and Crews, 1985], the subtasks we see in machine learning "general" benchmarks are not

axiomatic — they were not actively designed as abstractions of any meaningful general function or sub-function, and are often not systematically curated in nature. Instead, what we find in the "general" benchmarks are sets of categories reminiscent of the inconsistent hodgepodge of rooms that Grover visits in the "Everything in the Whole Wide World Museum".

### 4.1.1 Arbitrarily Selected Tasks and Collections

Concerns of this sort were raised during early benchmark development in the field. UCI [Dua and Graff, 2017] was one of the first foci of shared tasks in machine learning development and contains datasets which pertain to a collection of individual subtasks, with community benchmarking practice coming to focus over time on the Iris, Adult, Wine, and Breast Cancer classification datasets. Kiri Wagstaff voiced concern for a poor logic in the selection of this combination of subtasks. She noted that none of tasks included in this collection represent any reasonable proxy or abstraction of real-world problems even scientists in the respective fields that produced each dataset would care about. She notes, "Legions of (ML) researchers have chased after the best iris or mushroom classifier. Yet this flurry of effort does not seem to have had any impact on the fields of botany or mycology. Do scientists in these disciplines even need such a classifier? Do they publish about this subject in their journals?" [Wagstaff, 2012, p.2].

The classes in ImageNet are also seemingly arbitrary, ranging from specific dog breeds to high-level notions like "New Zealand beach" [Russakovsky et al., 2015]. ImageNet labels are derived from the 12 "subtrees" of the WordNet ontology [Miller, 1995]: mammal, bird, fish, reptile, amphibian, vehicle, furniture, musical instrument, geological formation, tool, flower, and fruit [Deng et al., 2009]. Despite being developed in another field, with another purpose entirely, the English-language WordNet hierarchy was adopted, nearly whole cloth, for use in structuring ImageNet. Little consideration was given to the impact of the inclusion of certain words in the task of image classification. The consequences of this oversight were made tangible by the work of Birhane and Prabhu [2021] and Crawford and Paglen [2019] which revealed the existence of a range of derogatory and offensive categories in the "person" subtree. The primary categorical consideration the ImageNet creators *did* discuss center around the inclusion of certain categories to ensure consistency with the categories in a previously established PASCAL benchmark [Russakovsky et al., 2015].

Similarly, the subtasks of the GLUE benchmark are not much more carefully selected. The benchmark was initially designed as a suite with a stated goal to "spur development of generalizable NLU systems" [Wang et al., 2019a, p.1], embodying the idea that a system capable of using linguistic knowledge to understand language input would be able to do so across a variety of tasks. The included tasks were curated following a set of 30 proposals sourced from with an informal survey of colleagues in the NLP community [Wang et al., 2019a], and thus only really represent a collection of tasks authors from the NLP community perceived as interesting problems at the time. To filter through proposals, authors mainly rely on practical heuristics like "licensing issues, complex formats, and insufficient headroom" [*Ibid.* p. 5] before referring to high level criteria to filter down to the final eight or nine included tasks. As a result, the collection of tasks included in the final benchmark neither systematically map out a range of specific linguistic skills required for understanding (lacking in particular any exploration of pragmatics [Qiu et al., 2020], or the proper handling of negation [Ettinger, 2020]) nor present a truly varied range of ways to deploy linguistic knowledge in comprehension. While such a collection can effectively demonstrate a model's ability to generalize performance across the included tasks, this ability is not equivalent to general language understanding, nor is it evidence that the solution to any task actually involves understanding the language in a sense that linguists might recognize.

### 4.1.2 Critical Misunderstandings of Domain Knowledge and Application Problem Space

GLUE and SuperGLUE benchmarks combine linguistic competence (the ability to model a linguistic system) with general commonsense and world reasoning as if they were equivalently scoped problems. In reference to a diagnostic component of the benchmark, authors state that "this dataset is designed to highlight common challenges, such as the use of world knowledge and logical operators, that we expect models must handle to robustly solve the tasks" [Wang et al., 2019a, p.1]. Thus a benchmark designed to test for generalizability across different language understanding tasks comes to subsume not only the task of building up linguistic competence (e.g. logical operators) in the language in question (English) but also the ability to acquire and deploy world knowledge. It is well established

that natural language understanding requires both linguistic processing and reasoning over the combination of the linguistic signal, communicative common ground, and world knowledge [Hunter et al., 2018], but while linguistic knowledge is relatively self-contained and reusable across different textual domains, world knowledge is open-ended. Conflating these two abilities in a benchmark that is easily (mis)interpreted as representing a much more general, flexible, and robust set of capabilities than it possesses, and thus inappropriately presented as comparable to the "human ability to understand language" [Wang et al., 2019a].

Furthermore, language understanding relies not only on linguistic competence but also world knowledge, commonsense reasoning, and the ability to model the interlocutor's state of mind [Reddy, 1979, Clark, 1996], none of which can be thoroughly tested through text-only tasks, such as GLUE. Several researchers have raised the need to establish effective physical and social grounding as part of the process of moving towards robust and effective natural language understanding, warning against text-only learning as a limited approach [Bisk et al., 2020, Zellers et al., 2021]. Bender and Koller [2020] additionally mention the tendency of machine learning researchers to misinterpret certain benchmarks as capturing the model's ability to decipher *meaning* in language, arguing that benchmarks need to be constructed with care if they are to show evidence of "understanding" as opposed to merely the ability to manipulate linguistic form sufficiently to pass the test.

## 4.2 De-contextualized Data and Performance Reporting

In "The Hall of Very, Very Light Things", Grover finds a big rock and declares, "There has been some mistake! This big rock is not light," before deciding to move it to the "The Hall of Very, Very Heavy Things". However, such judgements are ultimately relative — that rock is certainly much less heavy than the trailer truck in the latter room, and could be considered light in comparison. Nothing about the museum is neutrally determined.

In this section, we explore one of the features that can lead researchers to mistakenly construe a benchmark as "general", namely the de-contextualization of its component tasks and datasets. No dataset is neutral and there are inherent limits to what a benchmark can tell us — in fact, data benchmarks are closed and inherently subjective, localized constructions. If anything, the claim to generality will often act as cover, allowing those developing the benchmarks to escape the responsibility of reporting details of these limitations. Part of the challenge of addressing this lack of context is proper documentation for these datasets, which is often underdeveloped [Gebru et al., 2020, Bender and Friedman, 2018] and the devaluation of the data work [Paullada et al., 2020, Sambasivan et al., 2021, Hutchinson et al., 2020, Jo and Gebru, 2020].

### 4.2.1 Limited Scope

Even large datasets like ImageNet are ultimately closed systems, limited in their coverage of non-Western contexts and temporally bounded. Torralba and Efros [2011] demonstrate how images from the same class but different datasets are often distinguishable and embody a very specific style of capturing some segment of the real world. Specifically, a well-critiqued limitation of ImageNet is that the objects tend to be centered within the images, which does not reflect how "natural" images actually appear [Barbu et al., 2019]. Additionally, it's not clear that increasing the size of these benchmarks to capture omitted content is even practically feasible in many cases — ImageNet authors note the trade-off between annotation quality and size, remarking that "the scale [is] already imposing limits on the manual annotations that are feasible to obtain" [Russakovsky et al., 2015, p.34]. There are thus likely serious practical limits to making such benchmarks larger.

The notion of task diversity, not size, is repeated quite frequently by GLUE authors [Wang et al., 2019a, p. 1,2,3,5] as a main differentiator from predecessors' benchmarks, SentEval [Conneau and Kiela, 2018] and decaNLP [McCann et al., 2018]. However, despite the expressed desire for the benchmark to include a diverse range of tasks that "cover a diverse range of text genres, dataset sizes, and degrees of difficulty" [Wang et al., 2019a, p. 1], compared to human linguistic activity, the GLUE tasks are hardly diverse: two single-sentence tasks, two similarity and paraphrase tasks, and four inference tasks. As noted in the development of a subsequent iteration of the benchmark, SuperGLUE, "task formats in GLUE are limited to sentence- and sentence-pair classification"[Wang et al., 2019b, p. 1], requiring a subsequent expansion to include coreference resolution and question answering (QA) task formats in SuperGLUE. SuperGLUE includes four question answering tasks, two inference tasks, one word sense disambiguation task and one coreference task.

### 4.2.2 Benchmark Subjectivity

All datasets come with an embedded perspective — there is no neutral or universal dataset [Haraway, 1988, Stitzlein, 2004, Gebru, 2020, Denton et al., 2021, Scheuerman et al., 2021]. To present a dataset, inherently both political and value-laden, as a completely neutral scientific artifact is irresponsible. In benchmarks promoted to assess general capabilities, such as ImageNet and GLUE, such politics remain unacknowledged — undiscussed and hidden for the sake of maintaining the claim to broad relevance. However, denying DRdeletethe existence of unacknowledged context does not make it disappear. In fact, this distorted data lens is often not limited in an arbitrary way, but limited in a way that hurts certain groups of people — those without the power to define the data themselves. For example, people who hold transgender and gender-nonconforming gender identities and non-white, non-Western racial identities are underrepresented in mainstream face datasets [Merler et al., 2019], and images of members of these communities are often tagged with racial or ethnic slurs, even on large general use datasets such as MIT Tiny Images or ImageNet [Crawford and Paglen, 2019, Gehl et al., 2017, Birhane and Prabhu, 2021].

GLUE and SuperGLUE target one specific language (i.e. English), not "language" in the abstract. This very specific subset of American English text in the presence of annotation artifacts [Gururangan et al., 2018] results in inherently subjective outcomes [Waseem et al., 2020]. [3]

The ImageNet creators did attempt to diversify their dataset by translating image queries into other languages, including Chinese, Spanish, Dutch and Italian to source representations for the final English-labeled categories [Deng et al., 2009]. However, when analyzing country level geo-location data for the 2011 version of the dataset, 45% of the images were found to be sourced from the US, and over 60% from a small selection of Western countries in North America and Europe [Shankar et al., 2017]. In fact, only 1% and 1.2% of images are from China and India respectively, despite the fact that those countries are the most populous countries on the planet [Shankar et al., 2017]. This lack of geo-diversity of sources manifests in poor dataset representation, as the dataset becomes visually anchored to a specific dominant cultural context. de Vries et al. [2019] found that if ImageNet was developed using the results of Hindi online image queries, rather than English ones, the result would be a dataset that looks drastically different, embodying completely new visual representations of not only socially defined concepts, such as "wedding", but also everyday objects, such as "spice".

## 4.3 Inappropriate Community Use

When Grover arrived at the museum, he was excited and inspired by its claim to contain every object in the "Whole Wide World" — however, once he enters, his focus is only redirected to whatever is selected to be highlighted in the museum. Despite his initial interest being motivated by the pursuit of generality, it is only really the subset of consciously-included objects that actually captures his attention.

When individual dataset creators and communities overstate the generality of these benchmark datasets, they elevate them to the status of a target the entire field should be aiming for. This benchmark focus can escalate to the point of researchers falling into the trap of uncritically chasing algorithmic improvement as measured by these datasets, losing sight of the potential for performance mismatch with the real world or more relevant problem formulations.

### 4.3.1 Limits of Competitive Testing

Chasing "state of the art" (SOTA) performance is a very peculiar way of doing science — one that focuses on empirical and incremental work rather than hypothesis-based scientific inquiry [Hooker, 1995]. In 1995, Lorenza Saitta criticized benchmark chasing as "allow[ing] researchers to publish dull papers that proposed small variations of existing supervised learning algorithms and reported their small-but-significant incremental performance improvements in comparison studies" [Radin, 2017, p. 61]. Thomas and Uminsky go so far as to present metric chasing as an ethical issue, stating that "overemphasizing metrics leads to manipulation, gaming, a myopic focus on short-term goals, and other unexpected negative consequences" [Thomas and Uminsky, 2020, p. 1].

---

[3]Waseem et al. [2020] are discussing NLP datasets and modeling in general, not GLUE, SuperGLUE or other similar benchmarks specifically. However, their general points apply: any given dataset represents the embodied viewpoint of its authors and annotators, and furthermore, datasets constructed without attention to whose viewpoints are being represented will likely over-represent hegemonic ones. See also Bender et al. [2021].

Furthermore, Kiri Wagstaff points out some inherent flaws of the aggregate performance evaluation format for many "general" benchmarks and leader boards. She notes that "80% accuracy on Iris classification might be sufficient for the botany world, but to classify as poisonous or edible a mushroom you intend to ingest, perhaps 99% (or higher) accuracy is required. The assumption of cross-domain comparability is a mirage created by the application of metrics that have the same range, but not the same meaning. Suites of experiments are often summarized by the average accuracy across all datasets. This tells us nothing at all useful about generalization or impact, since the meaning of an x% improvement may be very different for different data sets (or classes)." [Wagstaff, 2012, p.3] Wagstaff here calls out the fact that the task is not just poorly formulated but also likely measured inappropriately, in a way that makes the results difficult to interpret meaningfully.

Despite these issues, both GLUE and ImageNet authors situate the benchmark datasets within designed competitive environments. GLUE authors cite "the leaderboard and error analysis toolkit" as a differentiating factor [Wang et al., 2019a, p.3], while ImageNet authors detail the format of "an annual competition and corresponding workshop" [Russakovsky et al., 2015, p.1,49].

### 4.3.2 Redirection of Focus for the Field

Benchmarks have always been historically influential, and are often created intentionally to incentivize interest in certain research topics. The National Institute of Standards and Technology (NIST) invested in excess of $6.5 million for the FERET facial recognition challenge from 1993 to 1998, in order to encourage research participation on a technology of interest to the sponsoring agency — the Department of Defense [Phillips et al., 2000]. The Netflix Prize, which ran from 2006 to 2009, was launched by Netflix, an online DVD-rental and video streaming service, in order to drive innovation for the development of collaborative filtering algorithms that improved the quality of automated movie recommendations [Bennett et al., 2007]. However, there is a notable difference between such competitions and those involving what are presented as "general" benchmarks — namely, the PASCAL VOC challenge running from 2005 to 2012, the ImageNet Large Scale Visual Recognition Challenge (ILSVRC) running annually from 2010 to 2017, and the GLUE/SuperGLUE leaderboard, which has been running since 2018. The focus of competitions involving "general" benchmarks are much further detached from grounded applications, with tasks more vaguely defined. As a result, these competitions become interpreted as significant markers of success across a variety of relevant sub-domains, and their popularity grows accordingly as that competition is elevated to represent the ultimate showcase of algorithmic performance.

Benchmarks also influence the nature of the dominant algorithmic approaches attempted. In the 1960s, chess caused the AI community to hyper-focus on deep-tree searching and the minimax algorithms, which were most effective on improving game performance. Both of these methods came to dominate the algorithmic development of this time, resulting in the neglect of alternate problems and approaches [Ensmenger, 2012]. Dotan and Milli [2020] note how trends for algorithmic development in ML are driven by performance on particular benchmarks, and how these trends result in shifts to the entire research program of the field. "General" benchmarks in particular often inspire resource-intensive models: both through presenting very large training sets for models to consume and by conflating size (of both training set and thus model) with generality. The current deep learning movement would not have been possible without ImageNet [Alom et al., 2018]. In a similar way, the success of BERT models, a breakthrough of language model development, was first effectively demonstrated using GLUE [Devlin et al., 2019].

### 4.3.3 Justification for Practical out of Context or Unsafe Applications

General benchmarks also get mentioned in marketing copy for commercial machine learning products, with performance on the benchmark presented as evidence of real-world technical achievement. This context is when the significance of benchmarks is most severely distorted, when performance on benchmarks is not just the tool for algorithmic selection, but actually presented as some reliable marker of expected model achievement in deployment. For instance, in January 2021, Microsoft claimed "DeBERTa surpassing human performance on SuperGLUE marks an important milestone toward general AI." [He et al., 2021]. ImageNet is equally elevated as an indicator of commercial model success — so much so that Baidu cheated on the benchmark in order to inflate product performance claims in marketing [Simonite, 2015].

Poor performance on minority subgroups could possibly be disguised in aggregate and decontextualized "general" benchmarks to justify deployment. We already know from examples of deployed, commercial products in facial recognition [Buolamwini and Gebru, 2018] that unbalanced representation impacts performance on certain groups over others, though this disproportionate performance is often hidden in aggregate performance measures on what turn out to be biased benchmarks [Merler et al., 2019, Raji and Buolamwini, 2019].

## 5   Alternative Roles for Benchmarking and Alternative Evaluation Methods

In preceding sections, we have explored the ways in which "general" benchmarks fail to serve as effective measures of progress in machine learning. Now we ask: What can be done instead? This is not a question of "fixing", "improving" or "expanding" the current "general" benchmarks — after all, the solution to addressing the issues inherent to the "Everything in the Whole Wide World" museum is not to add more rooms. The presentation of a data benchmark as being able to exist independent of context, scope and specificity is itself a false premise for machine learning evaluation.

Instead, we note two paths forward. First, we argue that benchmarks should be developed, presented and understood as intended — to evaluate concrete, well-scoped and contextualized tasks. Second, when probing for more broad model objectives, behaviors and capabilities, we propose that the field explore alternative evaluation methods. The following are some starting points on evaluation techniques to explore further in machine learning as an alternative to benchmarking (further details in Appendix A):

- *The systematic development of test items* (see Appendix A.1), in the form of testsuites [e.g. Lehmann et al., 1996, Ribeiro et al., 2020], audits [e.g. Buolamwini and Gebru, 2018], and adversarial testing [e.g. Ettinger et al., 2017, Niven and Kao, 2019]. These techniques can help map out which aspects of the problem space remain challenging and/or check for the potential for harms coming from system biases.

- *System output analysis / behavioural testing* (see Appendix A.2), including error analysis [e.g. van Miltenburg et al., 2021], disaggregated analysis [e.g. Raji and Buolamwini, 2019], and counterfactual analysis [e.g. Garg et al., 2019, Hutchinson et al., 2020]. These techniques can reveal the kinds of failure modes a system might produce, and potentially their causes.

- *Ablation testing* (see Appendix A.3) [e.g. Newell, 1975] allows researchers insight into the specific contributions of different system components.

- *Techniques for analyzing model properties* (see Appendix A.4) that are orthogonal to system outputs such as profiling energy consumption [e.g. Henderson et al., 2020, Schwartz et al., 2020], memory requirements [e.g. Ethayarajh and Jurafsky, 2020], and stability in the face of perturbations to training data [e.g. Sculley et al., 2018].

## 6   Conclusion

The situation with Grover and the museum's claims are clearly ridiculous — yet in machine learning, we follow the exact same logical fallacies to justify the elevation of a select number of benchmarks operating as general benchmarks for the field. However, there is no dataset that will be able to capture the full complexity of the details of existence, in the same way that there can be no museum to contain the full catalog of everything in the whole wide world. Open-world, universal and neutral datasets don't exist, and current methods of benchmarking do not offer meaningful measures of general capabilities.

The effective development of benchmarks is critical to progress in machine learning, but what makes a benchmark effective is not the strength of its arbitrary and false claim to "generality" but its effectiveness in how it helps us understand as researchers how certain systems work — and how they don't. Benchmarking, appropriately deployed, is not about winning a contest but more about surveying a landscape — the more we can re-frame, contextualize and appropriately scope these datasets, the more useful they will become as an informative dimension to more impactful algorithmic development and alternative evaluation methods (see Appendix A). Given the alternative roles and interpretations for evaluation we could explore, it is essential that we move quickly beyond the narrow-yet-totalizing lens of the "Everything in the Whole Wide World" benchmarks.

# 7 Acknowledgments

The authors thank Sam Bowman, Julian Michael, and Ludwig Schmidt for comments on earlier drafts of this work, as well as the organizers and attendees of the NeurIPS 2020 Workshop on ML Retrospectives, Surveys, and Meta-Analyses for providing an early venue for this work. We also thank the reviewers for their comments.

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
