# A Appendix A: Details of Alternative Evaluation Methods

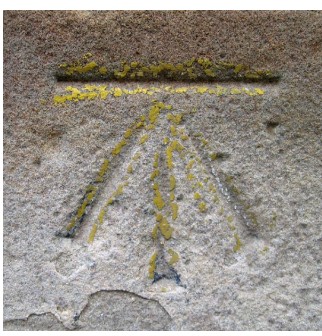

Figure 1: Photo of a benchmark in Edinburgh, by Jeremy Atherton, used without modification according to license CC-BY-SA-2.5

The etymology for the term "benchmark" refers to a mark that was added to buildings to indicate the position of a surveyor's bench (see Fig. 1), itself a tool for creating a level surface on which to put a leveling rod, used in the process of surveying. This etymological source contrasts with the use of benchmarks for charting the state of the art: surveyors are not in the business of measuring the furthest anyone has gone along some particular trail but rather in understanding the shape of the landscape and how it has changed over time [Kahmen and Faig, 1988].

We can take further inspiration from the field of surveying, as we think about how the measurements we take relate to the terrain we wish to understand: "A surveyor is not only charged with providing results derived from [their] measurements, but also has to give an indication of the quality and reliability of these. This requires a clear understanding of the functional and stochastic relationships between measured quantities and derived results, as well as a solid understanding of the external factors that influence the measurements." [Kahmen and Faig, 1988, p.1].

We describe in this appendix other methodologies that can move us towards the perhaps more worthy goal of filling in our picture of the landscape. In turn, we provide an overview of alternative, underexplored evaluation techniques including testsuites, audits and adversarial testing; system output analysis; ablation testing; and analysis of model properties.

## A.1 Testsuites, Audits and Adversarial Testing

Typical benchmark evaluation datasets are sampled from some larger dataset (e.g. via a train/test split) such that the frequency distribution of test item *types* in the test data is influenced by their distribution in that underlying dataset. In contrast, testsuites and audits specifically design their test sets to map out some space of test item types and evaluate systems in terms of the extent to which they can handle them.[4] The testsuite-based approach has a long history in NLP, with notable early publications including Lehmann et al. [1996] and recent work such as Ribeiro et al. [2020].

There are various approaches to the design of such test suites. A more audit-like methodology, exemplified by Buolamwini and Gebru [2018], creates test sets balanced for sensitive categories so as to be able to test for differential performance across those categories. On the other hand, explicit adversarial testing seeks to explore the edges of a system's competence by finding minimally contrasting pairs of examples where the system being tested succeeds on one member of the pair and fails on the other [Ettinger et al., 2017].[5] Importantly, these evaluation approaches are designed around diagnosing particular areas of system failure: the point of the tests isn't to show which systems can "solve" them but to understand which aspects of the problem space remain challenging and represent the operational limits in deployment.

## A.2 System Output Analysis

System output analysis is another way to explore the system output in detail. This can take the form of error analysis, disaggregated analysis, and counterfactual analysis.

**Error analysis** Error analysis involves the detailed analysis of system errors, by either mechanically or manually inspecting system inputs. Mechanical analysis can include simple techniques such as developing a confusion matrix in labeling tasks. A confusion matrix compares expected labels to system output labels and provides a summary of which categories are most reliably labeled and which

---

[4]A notable exception to the trend of benchmarks to not include test suites is GLUE, which includes among its component tests a testsuite mapping out various linguistic constructions in English [Wang et al., 2019a].

[5]Adversarial testing also includes work like Niven and Kao [2019] that discovers what kind of spurious cues systems are leveraging to effectively "cheat" on a particular benchmark and creates alternate versions of the testsets that neutralize those cues.

are most frequently confused for each other. (This is an old technique, initially used in the study of human phonetic perception [Miller and Nicely, 1955] and related to even earlier work comparing medical diagnostics when lacking access to ground truth [Yerushalmy, 1947].) Other mechanical analyses include looking at errors by easily automatically measurable properties of system input such as sentence length or presence or number of out of vocabulary items.

More detailed error analysis digs into the specific system inputs to look for patterns that can't necessarily be measured automatically and might not be modeled in any way by the system being evaluated. For example, error analysis of a sentiment analysis system might find that it is frequently tripped up by sarcasm or in a simpler case by sentences with phenomena such as coordination or subordinate clauses. Error analysis of a machine translation (MT) system might find that it frequently fails on examples with negation [Wetzel and Bond, 2012, Fancellu and Webber, 2015, Hossain et al., 2020]. Error analysis can turn up important problems that don't have a large effect on the metric. For example, metrics for MT systems (including but not limited to BLEU) are also not good at measuring the impact of negation errors [Hossain et al., 2020].

**Disaggregated analysis** Disaggregated analysis can reveal disparate patterns of performance that may not be visible through aggregate metrics alone. This method has been leveraged within the ground-breaking audit of facial analysis systems, performed by Buolamwini and Gebru [2018], that evaluated performance across unitary and intersectional subgroups defined by gender and Fitzpatrick skin type. This analysis revealed significant disparities in model performance — with darker-skinned female subjects experiencing the highest error rates — that was not visible through examination of aggregate performance metrics alone. The method of disaggregated analysis has since been adopted by a myriad of auditing and evaluation works [e.g. Raji and Buolamwini, 2019] and integrated into frameworks of standardized model reporting [Mitchell et al., 2019]. Following these works, we encourage researchers to report performance metrics on socially salient slices of their dataset, in addition to the full test set.

**Counterfactual analysis** Counterfactual analysis is another technique of model evaluation and assessment that has gained in popularity in recent years. At a high level, these methods evaluate how a model's output changes in response to a counterfactual change in the input. This method has been leveraged for fairness-informed analysis of natural language processing systems by comparing model performance on paired inputs that differ only in a reference to a sensitive identity group [Garg et al., 2019, Hutchinson et al., 2020]. While both counterfactual analysis and disaggregated analysis have been leveraged to disparities in model performance for different sensitive groups, counterfactual analysis can additionally provide insight into causal mechanisms underlying particular patterns in performance. Counterfactual analysis can also be leveraged to assess model robustness to small distribution shifts [Christensen and Connault, 2019].

The results of system output analysis tend to be rich and detailed and not amenable for quick cross-system comparison. But this is a feature, in our view, and not a bug: the goal, after all, is not anoint one system the winner (until some new system claims that spot), but rather to understand how aspects of system design map onto different aspects of the problem space so as to inform the next iteration of system development.

### A.3   Ablation Testing

Another well-established technique for understanding system performance is ablation testing. Ablation testing, as named by Newell [1975] by analogy with similar studies in biology, involves isolating the contributions of different system components by removing them, one by one, and evaluating the modified system. In statistical NLP prior to deep learning approaches, ablation testing was commonly applied to different feature sets to explore the extent to which systems were using information captured by different aspects of input representation. Ablation testing can also be performed on subsets of training data to explore the effect of e.g. in-domain v. out-of-domain training data or on components of system architecture, to the extent that these can be removed without completely disabling the system. Heinzerling [2019], in a discussion inspired by Niven and Kao [2019], suggests various data ablations that can be applied to test data as well, to investigate what cues a system might be relying on.

## A.4  Analysis of Model Properties

System performance on test items (in aggregate as in test sets in standard benchmarks or in detail as in error analysis or test suites or audits) is only one facet of systems to consider, especially when trying to gauge which approaches are most feasible for practical applications. Other dimensions include energy consumption (both for development and testing) [Strubell et al., 2019, Henderson et al., 2020, Schwartz et al., 2020, Ethayarajh and Jurafsky, 2020]; memory and compute requirements, which may be more or less constrained depending on the deployment context [Ethayarajh and Jurafsky, 2020, Dodge et al., 2019]; and stability in the face of perturbations to the training data [Sculley et al., 2018]. This latter is especially important for systems that need to be continually retrained in order to handle changing input data, such as named entity recognition system that need to keep up with different public figures of note.