# OpenReview forum: "AI and the Everything in the Whole Wide World Benchmark"
_NeurIPS.cc/2021/Track/Datasets_and_Benchmarks/Round2 — NeurIPS 2021 Datasets and Benchmarks Track (Round 2)_

### Official Review · Reviewer_JynM · 2021-09-19
**A compelling argument against "general" monolithic benchmarks in vision and NLP**

**Rating:** 7
**Confidence:** 3
**Correctness:** The claims in the submission seem cor…
**Clarity:** This paper was a good read as it is v…

**Strengths:**

The paper provides many very compelling arguments that monolithic benchmarks should not be so heavily relied upon as measures of progress within subfields of machine learning, and highlights many of the problems with chasing “SoTA” performance on these benchmarks. These contributions are valuable to the machine learning community and should be taken seriously. The authors argue that it’s actually impossible for specific benchmarks to act as perfect proxies for general intelligence (more specifically, as it pertains to vision and language understanding problems), and it's clear that this perspective is very often neglected in the community - particularly, I think, as it pertains more concretely to the publication process.

The two benchmarks which are at the center of the discussion are two of the most widely adopted means of evaluation for vision and NLP, which are themselves two of the most widely studied domains in machine learning - which makes the scope of the argument in the paper immediately relevant to a very large portion of the machine learning community, and likely indirectly relevant in various ways to the entire machine learning community as new benchmarks are developed for new domains. The authors provide a number of novel arguments while highlighting arguments and analyses made in other works which support their claims. I would like to see the computer vision and NLP communities take these arguments seriously, as I think they are a valuable contribution toward making more meaningful progress in these fields beyond chasing the state-of-the-art.

Note that while the study provides no quantitative analysis or concrete dataset or benchmark contribution (which would probably be orthogonal to the arguments by the authors anyway), the work is within the scope of the Datasets and Benchmarks track (i.e., “identifying significant problems with existing datasets and their use”).


**Weaknesses:**

While I agree with virtually all of the arguments presented in the paper, after an initial read, I was left somewhat unsure as to what actions an individual researcher proposing a new method could take to mitigate reliance on popular benchmarks falsely claiming generality. It is often the case that researchers are systematically pressured into relying on benchmarks such as ImageNet and GLUE as part of the publication process.

**Additional Feedback:**

-	Please use the latex template designed for the Datasets and Benchmarks track (page 1: “NeurIPS 2020 Workshop: ML Retrospectives, Surveys & Meta-analyses (ML-RSA)” seems to be a holdover from a previous submission).
-	Missing space in section 3.2 “well-defined kind of generalizability.Furthermore,”
-	Same in 4.1.1. “are not much more carefully selected.The benchmark”, “(lacking in particular any exploration of pragmatics[Qiu et al., 2020], or the proper handling of negation[Ettinger, 2020])”


**Documentation:**

Not applicable, so none provided.

**Ethics:**

No ethical review required.

**Relation To Prior Work:**

The paper clearly identifies and discusses prior work analyzing problems in benchmarking NLP and vision while making broader (novel) points about the use of “general” benchmarks, which separates this paper from prior work.

**Summary And Contributions:**

This position paper argues that large-scale “general” benchmarks are in fact deceptively limited and criticizes the widespread reliance on such benchmarks as monolithic measures of progress in their respective domains. The paper centers its argument on the ImageNet and GLUE benchmarks. The paper begins with background on research striving for artificial general intelligence and of benchmarking more broadly (which can be traced back to benchmarking computer performance) and goes on to discuss how benchmarking in machine learning began as standardized ways of evaluating performance on narrowly scoped tasks. The authors then argue that benchmarks for general-purpose language understanding and general-purpose object recognition are fundamentally flawed because they are no longer narrowly scoped and falsely claim to be real-world-esque proxies for generality beyond the data provided in said benchmark (which is, by definition, fixed). These claims are supported by arguments that any fixed dataset contains subjective bias, which has been previously shown to be true of the benchmarks in question – ImageNet and GLUE. Many other problems surrounding ImageNet and GLUE are discussed, including limitations in task design, problems with task/dataset collection, a lack of domain knowledge, a lack of context regarding tasks/datasets, limitations in scope, misuse by the community, inappropriate SoTA-chasing, and the problem of influencing the direction of the field away from grounded applications/the popularity of certain types of algorithms.

---

> ### Author Response · Authors · 2021-09-28
> **Thanks for this response!**
>
> Thanks for this response! We have the following comments to add below:
>
> >  unsure as to what actions an individual researcher proposing a new method could take to mitigate reliance on popular benchmarks falsely claiming generality.
>
> Our position is that the performance on current dominant benchmarks such as ImageNet and GLUE should be more appropriately scoped.
>
> We provide several recommendations for improved evaluation practices in Appendix A. We plan to tighten the integration of the Appendix with Sec 4 to make those more apparent. Beyond improving their own research practice, we hope that individual researchers will also keep these points in mind while serving as reviewers. We will raise that suggestion and other clear action items in the revised paper.
>
> We agree that current publication incentives have contributed to many of the named issues in our paper, but are hopeful that by mapping out these arguments, we can motivate a shift in how we think of the role of such benchmarks in our field as well as prompt some further reflection on how we present and discuss such benchmarks.
>
> > Additional feedback
>
> Thank you for catching these. We will certainly fix these (and other typos).

---

### Official Review · Reviewer_RQ5L · 2021-09-22
**The Fallacy of Benchmarks for General Artificial Intelligence**

**Rating:** 7
**Confidence:** 4

**Strengths:**

The historical background is very well done and very informative. One can tell it is very well researched, and everybody can learn something new from this paper.

The intervention of construct validity is important, and I think it is something that will hopefully be studied more on the AI + society side.

Very good job with all the suggestions for improving benchmarks. :)


**Weaknesses:**

The first paragraph should be more tied together.
"He opens the door, only to find himself in the outside world." -- How do the authors of this paper interpret this?
"As a children’s story, Grover’s described situation is meant to be absurd." -- The word absurd doesn't 100% tie in to the afore sentences.
"However, in this paper, we discuss how a similar faulty logic is inherent to recent trends in artificial intelligence" -- You never stated what the logic is.
In sum, there needs to be more explication and tying together of these concepts.

To strengthen the paper, be careful about making claims that a) have no literature you can currently cite or b) are not supported through argumentation.

**Additional Feedback:**

Overall, the paper is really good, and the recommendations for moving forward with benchmarks provide a lot of pizzazz. Where I'm not 100% sold is the idea that the mentioned benchmarks are "general purpose" in the sense of the overarching goal of AGI rather than general purpose in a narrow AI specialty, even the authors mention that AGI is underspecified. For example, is computer vision one piece of AGI, or do we get to an intelligent agent that can understand its environment another way? Another reason I'm not sold is that the authors could have made the claim that the promise of current benchmarks is overhyped and not rooted in practical task specifications without making the claim that these are general purpose benchmarks and tying them to AGI. The AGI conversation somewhat seems to fall apart very easily (to me because I also read a lot on AGI although I don't care to touch it with a ten foot pole because it raises more questions than it answers) when considering benchmarks for "narrow AI".

**Clarity:**

The paper is not often clear at times. The introduction has a lot of jargon which is not immediately explicated although I respect the page count limitations.

In 2.1 paragraph 2, I am unsure if the authors are making the claim that the AI and ML fields collectively have the goal to move toward AGI because the following sentence states it is a subset of the community. If there is a collective goal, I would cite literature supporting this claim, so that it does not appear as a broad sweeping statement.

If I'm not mistaken, the argument in the paper asserts that more progress can be made if, among other things (e.g., see section 4 writ large), we stop glamorizing the false promises of general purpose benchmarks, which will allow AI researchers to understand where we really are and potentially how we should proceed given this knowledge of our current state of affairs.

**Correctness:**

My unsolicited opinions/thoughts:
Computer vision/NLP benchmarks are being tagged as general benchmarks when really the AI community would probably consider that narrow AI. So then, it is really a "general" intelligence in one "cognitive" area which I understand is somewhat wonky to reconcile.

I think it's easier to say that AI researchers should avoid hyping up their benchmark as moving towards a "generalized" intelligence in a specific area (e.g., computer vision) than it is to convince them to stop doing it. In reality, A LOT of structures actually support AI researchers providing more hype than outcome. In particular, a) professors have to fund their research labs, b) industry researchers who have to justify company expenditure, and c) the U.S. government wielding AI in a scientific battle with China. Therefore, there is a lot of incentive to make false promises or put up facades.

I think the philosophy of artificial intelligence from a machine learning researcher's perspective is actually richer than the sole presentation that the whole AI field thinks benchmarks get the field closer to general intelligence. For example, there are papers on how machine learning alone will not get us to general artificial intelligence alone, the misaligned representation of cognitive science in current machine learning that impacts general intelligence, and many researchers will state how dumb AI can actually be and how far away it is from general intelligence. I suppose there are equally more papers hyping up a facade of machine learning as better than it is, but I maintain that these are mostly a result of economic incentives. Even more so, I think there may be AI researchers for whom general intelligence is not a goal at all. Anyway, I mention all that to say that the artificial intelligence community is has varying opinions on this topic and there are varying opinions.

One criticism I have in general about everybody's claim that AGI is the goal of the AI community is that a lot of business would be happy with ML as it stands since it can be used for business analytics to make money, so who would be the stakeholders of AGI and what do they hope to achieve with AGI?

I know there is some discussion in the paper about general purpose NLP benchmarks, but some in the NLP community, particularly Dan Jurafsky, have expressed concerns about the state of benchmarks in NLP (e.g., inadequacy), so I don't think the NLP community has a universal opinion on NLP benchmarks being general purpose about as far as I can tell.

**Documentation:**

N/A

**Relation To Prior Work:**

This paper provides an original contribution by rethinking the way benchmarks are portrayed and how they are approached. It not only thinks through the fallacies of modern benchmarks, but provides ways to potentially improve the state of benchmarking

**Summary And Contributions:**

In "AI and the Everything in the Whole Wide World Benchmark", the authors assert that AI researchers should not advertise benchmarks as machine learning concepts that can move AI towards AGI nor should people buy into the idea that a benchmark can help achieve general intelligence. Starting with the concept of Grover and the Everything in the World Special, the authors support their thesis by providing a sociohistorical analysis of benchmarks in which the authors argue that the introduction of "general purpose benchmarks" that are a far cry from the original narrow AI task benchmarks with practical real-world tasks impede meaningful progress in the AI field. The authors' apparent purpose is demonstrate that benchmarks are not indicative of general artificial intelligence in order to convince AI researchers to change their framing in benchmarks as general purpose. The intended audience are artificial intelligence researchers at large, and the authors establish a relationship with the audience through their retelling of the history of AI and associated benchmarks as well as the Sesame Street concept. The authors' contribution is not only providing this intervention, but providing ways forward for the ML community.

---

> ### Author Response · Authors · 2021-09-28
> **We appreciate your thoughtful review!**
>
> We appreciate your thoughtful review. Our replies to the specific points raised are mentioned below:
>
> >  The first paragraph should be more tied together.
>
> We appreciate the feedback that the connection between the Grover story and what we see with “general” benchmarks can be made more explicit and will revise with that in mind. Particularly, we hope to clarify our understanding of the connection between the Grover story and our main point in this paper - the absurdity of the claim of a museum containing the contents of the whole world matches the absurdity of a claim that there can be a single benchmark that can play this role of assessing a representative range of "general" model capabilities.
>
>  In addition, we will look to see if there are terms we can revise around because there isn’t really room to define them in the introduction. Are there particular terms that struck you as jargon,  beyond Common Task Framework (CTF)? We hope to clarify such terms further in the text.
>
> > Computer vision/NLP benchmarks are being tagged as general benchmarks when really the AI community would probably consider that narrow AI.
>
> We are critiquing generality claims for CV or NLP benchmarks, because benchmarks in those areas are *explicitly* hyped as “general” (independent of sometimes being hyped as steps towards AGI). Even though a subset of the community may consider such applications “narrow AI”, the language of generality is explicitly being used by the dataset creators and the ML community when describing these benchmarks (for example, GLUE is named “*General* Language Understanding Evaluation”).
>
> That being said, this feedback helps us to see that we should further decenter any AGI discussion in the paper, since it is not essential to our point and indeed not the goal of everyone in the field, but just one interpretation of the motivation for developing and evaluating “general” performance. Overall, a critique of the AGI goal is a separate though related discussion to what we raise as an issue in this paper. Our main position is just to note that dataset creators are making certain types of claims that cannot be supported under the benchmarking paradigm, and that the goal of  AGI, as well as other types of generalization claims, motivate such inappropriate claims.
>
> Furthermore, in Sec 2.,1 we will do an editing pass to ensure that we are not ascribing such beliefs to all members of the field. Rather, we are calling out discourses which seem to be prominent in shaping the direction of the field, and pointing to the problematic benchmarking practices that result from this dominant rhetoric.
>
> Finally, we are quite curious to see Jurafsky’s remarks that you mention. Could you please clarify this with a citation?
>
> > In reality, A LOT of structures actually support AI researchers providing more hype than outcome.
>
> We definitely agree and we don't expect one paper to fix the many consequences of hype on the research practice in the field. However, we believe that it is really important to be having this conversation about benchmarks specifically, given their central role in shaping evaluation outcomes, whether as deployment criteria and to track notions of progress. Also, a paper like ours that concretely lays out some of the problems with current approaches can be helpful in motivating changes to incentive structures overall.

---

### Official Review · Reviewer_TV3q · 2021-09-22
**Well argued paper, with some reorganization suggested**

**Rating:** 7
**Confidence:** 4
**Correctness:** Yes
**Clarity:** Yes

**Strengths:**

The main strength is that for the points it makes, it does a very thorough job of supporting them. It's an important topic, and one that affects (nearly) anyone who works within ML research.

**Weaknesses:**

While the paper does a very good job outlining its core arguments about what is wrong with the current benchmarking paradigm, and it *does* include some suggestions in the appendix, at multiple times the lack of these suggestions in the main paper cause a feeling of "what would be an appropriate alternative?" The inclusion of this in the appendix makes me think it could be included in the main paper, especially if a camera ready has extra pages. However, a reorganization, where some suggestions are provided in each of 4.{1,2,3} would help the forward-looking part of the argument. Sometimes this may be somewhat straightforward based on the appendix, though in other cases it can be difficult. For example, a case study example for both ImageNet and GLUE in 4.1.1 suggesting fixes for the identified limitations would help.

**Additional Feedback:**

N/A

**Documentation:**

N/A

**Relation To Prior Work:**

Yes: see previous box(es).

**Summary And Contributions:**

This position paper argues that the standard evaluation practice of "benchmarking" within ML fields encourages over-generalization, loss of (scientific) focus, and broader harm. Its argument involves historical context, which helps situate and support its claims. The paper is very well written. The core contribution is a well-argued call to re-examine how ML fields handle evaluating progress.

---

> ### Author Response · Authors · 2021-09-28
> **Thank you, for your constructive reactions to our paper.**
>
> Thank you, for your constructive reactions to our paper. See below for our replies:
>
> >  "what would be an appropriate alternative?"
>
> We note two paths forward - to use benchmarking as intended to evaluate concrete and well scoped and contextualized tasks, or to explore alternative evaluation methods for assessing or probing for more broad model objectives, behaviors and capabilities. We can elaborate on this call to action in a future version of this paper.
>
> We appreciate the suggestion of moving our recommendations from the appendix into the main article. Given the space constraints, we could not do so for our submission. In a future version, we hope to add more pointers to the appendix in the main body of the text.
>
> > Mention specific updates to ImageNet and GLUE, fixes
>
> Our main point concerns the misalignment between benchmarking as an evaluation method and the claims that have been made on the basis of the benchmarks. We posit that there is no series of changes that can be made to a benchmark to enable it to support the claims of generality being made, due to the limitations articulated in Section 4. Thus, our suggestions are mostly around the presentation of the benchmark and properly scoping claims in this context. We hope ImageNet and GLUE can be informed by this work to adjust their claims on what performance on their benchmarks actually represent.
>
> As we mention in the Appendix, there exist other evaluation methods beyond benchmarking. Some of these methods - such as the behavioral method of test suites, for example [1] - are already increasingly common strategies for the evaluation of large models, with the hopes of characterizing the performance of more "general purpose" models through the lens of multiple assessments for specific, defined capabilities [2]. What we advocate for is moving away from an all in one benchmarking approach for the assessment of general capabilities so model performance claims can be better defined, contextualized and well-scoped.
>
> 1. Ribeiro, Marco Tulio, et al. "Beyond accuracy: Behavioral testing of NLP models with CheckList." arXiv preprint arXiv:2005.04118 (2020).
>
> 2. Rogers, Anna, Olga Kovaleva, and Anna Rumshisky. "A primer in bertology: What we know about how bert works." Transactions of the Association for Computational Linguistics 8 (2020): 842-866.

---

### Official Review · Reviewer_xcN4 · 2021-09-23
**History of the benchmarks we use today**

**Rating:** 5
**Confidence:** 3
**Clarity:** The paper is well written with helpfu…

**Strengths:**

* Extensive and insightful survey of the history of machine learning benchmarks and the dangers of applying them to increasingly general tasks.
* Examines the implications of benchmark design choices, such as the attempts to improve diversity of imagenet through query translation into other languages.


**Weaknesses:**

* The paper points out multiple problems, but doesn’t take a position.

Should the field move back to narrow and practical tasks for unsolved problems or continue to pursue more general objectives?

How should RecSys benchmarks make it simple for evaluation while being large enough to be representative?

Should an enhanced ImageNet improve diversity?  Along which dimensions?

How can benchmarks be realigned with real world tasks?

**Additional Feedback:**

If the field shouldn't benchmark general capabilities, what should it benchmark?

**Correctness:**

It is historically accurate as far as I can tell.


**Documentation:**

n/a

**Relation To Prior Work:**

It surveys benchmarking across multiple fields.

**Summary And Contributions:**

This position paper surveys the history of influential machine learning benchmarks, highlighting their limitations.  It notes that many benchmarks were influenced by the DARPA Common Task Framework, which focused on narrow practical tasks like ASR and MT, but has now evolved to include less practical and harder to measure general objectives.

---

> ### Author Response · Authors · 2021-09-28
> **Thank you for your comments!**
>
> Thank you, for your thoughtful comments and reactions to our paper. See below for our replies:
>
> > The paper doesn’t take a position
>
> We believe that we do take a position, though we hope to make this position more evidently articulated in a future version of the paper. In particular, it is our position that researchers in the field should not use performance on data benchmarks as evidence for general capabilities or progress towards such capabilities. We note throughout the paper that benchmarking as a method of evaluation is incompatible with such a vaguely-defined goal.
>
> > Should the field move back to narrow and practical tasks for unsolved problems or continue to pursue more general objectives?
>
> This question is outside of the scope of the main inquiry of this paper. In the paper, we do not hope to critique the objectives pursued - narrow and practical tasks are rather just the ideal tasks to instantiate meaningfully in benchmarks. More general objectives, if chosen to be pursued, need to be evaluated differently, as they are not appropriately represented in data benchmarks for the reasons discussed in the paper. If as a field we insist on the use of the benchmarking paradigm, then we need to understand the scope within which this evaluation method is most effective. This is the main perspective provided in this work.
>
> > How should RecSys benchmarks make it simple for evaluation while being large enough to be representative?
>
> We believe that this question is out of scope of the current inquiry. We specifically hope to critique benchmarks that explicitly make use of the rhetoric of being “general” purpose or representative of “general” capabilities. Our discussion examples are thus scoped to computer vision and natural language processing datasets that specifically make use of this rhetoric in their claims. Also, the notion of evaluation simplicity and representation is only addressed in the context of claims to “general” capabilities in performance, and not as a broad set of characteristics of benchmarks more broadly.
>
> > Should an enhanced ImageNet improve diversity? Along which dimensions?
>
> In this paper, our goal is not to critique the lack of diversity in ImageNet in order to offer points of improvement, but rather in order to directly reveal the limitations of a closed dataset like ImageNet to represent the open problem of general object recognition. Our point isn’t that these specific benchmarks fall short, but rather that there is a fundamental mismatch between the oft-stated claims of generality and the fundamental properties of benchmarks in evaluation practice.
>
> > How can benchmarks be realigned with real world tasks?
>
> Benchmarks can be aligned with real world tasks by investing in understanding the nuances of the task in the world and how the automated system relates to that task. A key part of this is in scoping the tasks to ones that can be modeled by datasets (as discussed in 4.1), but another element of this is crafting benchmarks with intent and incorporating feedback from domain experts.
>
> > If the field shouldn't benchmark general capabilities, what should it benchmark?
>
> We demonstrate in this paper that the benchmarking paradigm is in many ways incompatible with the assessment of broad general capabilities, but is rather an evaluation tool geared towards the assessment of more well-defined and scoped capacities. For those who have the goal of working on general capabilities, there is important science to be done in developing appropriate evaluation strategies. Our contribution is to show that the existing practice (using benchmarking for this purpose) does not suffice. If the field is committed to the benchmarking paradigm as the dominant mode of evaluating methods and measuring progress, then benchmarks should be deliberately designed, properly scoped and contextualized as originally intended, to concrete and well articulated problems.
>
> As we revise the paper for camera ready, we will take the above into consideration. It is very valuable to us to know in which ways the presentation was not clear.

---

### Official Review · Reviewer_AZHN · 2021-09-26
**Great, but improvements needed**

**Rating:** 6
**Confidence:** 5

**Strengths:**

The core argument, that benchmarks are an imperfect and epistemically limited way to track performance on (especially, but not only) open-world problems, is important, and the paper does a good job presenting it and summarizing both recent and historical literature in the area. In particular, I am pleased to see the concept of construct validity deployed to make this argument. The paper presents and organizes many ideas supporting this argument I had not previously considered or at least had not reduced to words as pointedly as this paper does.

Focusing on a small number of well-understood benchmarks accepted as "leaderboard problems" for open-world ML tasks (object recognition and natural language processing) provides a useful lens through which to bring the presented theory back "down to earth" and connect the arguments to the target audience of benchmark developers and users.

**Weaknesses:**

To me, the paper's biggest weakness is that it buys too much into exactly the current ML discourse/hype it means to debunk. This will be a theme across the specific weaknesses I describe below, and I think it can be avoided entirely with some small changes (again, described below).

First, the paper critiques the idea of benchmarks as valid evaluations of progress on general problems. But this is obvious, or at least should be - only the current ML hype makes it non-obvious (that doesn't make the critique unimportant! Only it makes the need for a precise strike at the core problem all the more critical). Benchmarks were always proxies for the performance criterion of interest. Even in the system performance benchmarks of the 1970-1980s, standard benchmarks were not "axiomatic" or descriptors of how a system would handle an untested workload (think, e.g., about the difficulty of interpreting performance benchmarks during Apple's recent transition from Intel processors to Apple-designed ARM-based processors). But they were better proxies for this than current ML benchmarks might be for the generalization problems of interest. So it's all about whether a benchmark is or isn't a good operationalization of the performance measure of interest. I think the paper could pretty easily reframe its critique of benchmarks in this language, claiming and then supporting the claim that existing benchmarks not only _aren't_ good operationalizations of the kind of generalization performance benchmark authors and leaderboard participants say they are, but _can't_ be for the reasons already offered. This provides a nice tie-in to the idea of construct validity used to make the argument already, so I think the subtle shift in language only strengthens the paper's argument and structure. It might be worth at least mentioning (briefly - 1 sentence, perhaps) the related management and political economy issues - why do we use certain benchmarks as proxies? Who chooses this and who benefits from those choices? Does the performance construct of interest make sense at a technical/construct validity level, or is it merely a construct of convenience (as opposed to an operationalization of convenience, as the paper already discusses). The argument at the very bottom of p5 is important, and deserves some expansion - since ML tasks are not necessarily related hierarchically, it is only by seeing the distinction between the benchmark and the performance goal of interest that we can even interpret what benchmark performance means.

Second, I am surprised to see an argument against the use of benchmarks for understanding progress towards general-purpose tasks which does not explicitly mention either the body of literature on generalization error and robustness or the body of literature on understanding epistemic error. Indeed, as far as I can see, neither the term "generalization" nor the term "epistemic error" appears in the paper. But controlling these is something we (at least ought to) teach students to do in month 1 of a machine learning class when we talk about overfitting and stopping criteria for training. Indeed, I kept expecting the paper to say something about whether benchmark performance leads to over-optimistic performance judgements about what happens when a model comes in contact with "real-world" data (and the related problem of creating/managing post-deployment "ground truth" or even just labeled data). It is particularly jarring not to see any discussion of concept drift or other forms of practical drift which are well studied. If anything is going to resonate with what I think the target audience of this paper is, I think it's this argument. A related argument surrounds what gets hidden by "headline numbers": accuracy on its own is a poor proxy for performance - the paper gets near this topic, discussing some recent work about bias and inequitable outcomes in specific cases, but without actually saying the important core idea: that high accuracy might belie highly unbalanced error rates for small classes or otherwise under-report performance for rare cases. Indeed, it would be useful to have one sentence somewhere that just indicates how ML tends toward the mean. This issue is particularly foregrounded for me in Section 4.1.3, where I think a critique of the concept on technical merits would support the already-strong critique offered on a philosophy-of-science basis (this does not diminish the validity of the proffered critique! But also: you don't want your audience to miss the trees for the forest).

Less importantly, I can't tell where the authors stand on the idea of general performance abstractly. Is the argument against benchmarks also an argument against general-purpose models and AGI? It could be, and perhaps the hesitancy to take this position is justifiable. But I think the paper needs to be clearer on this point. Either general performance is possible and benchmarks aren't telling us whether we're making progress on it (I _think_ this is the position, minus explicitly stating the first part?) or general performance is a misguided goal for the reasons stated in the paper (in which case, the paper should more clearly make that case). This may be a difficult suggestion to take action on, since the very concept of "general intelligence" is actually quite slippery for many of the reasons described in the paper. But I think distinguishing between these two cases will help the argument by focusing the criticism either on the goal or on the way the goal is operationalized. This fits into my overarching "theme" in that the task example of "natural language understanding" could be poorly represented by benchmarks like GLUE/SuperGLUE either because the benchmarks do a poor job of testing knowledge about the underlying world (as the paper mentions) or because the task of "understanding" is difficult to operationalize or because it's poorly formed as a problem. Does the paper accept this framing or is it critiquing it? How does the benchmark relate to other approaches that attempt to test this aspect of natural language models, like the Winograd Schema Challenge?

Minor but related to the specific point just made about NLP vs. NLU, the paper is less than clear about whether the relevant task is language _modeling_ or performance on some specified _task_ (I may not know enough about GLUE/SuperGLUE and possibly the answer is obvious if one does, but if so the relevant background needs to be presented!). These are not the same, and one does not imply the other automatically. For example, even if we accept a functional operationalization of "intelligence", a parser which identifies the parse/derivation tree of a sentence in Chomsky Normal Form cannot be said to be intelligent (although it might help as a component of a system that performs some NLP task well). I noted this issue specifically in the first paragraph of 4.1.2

An incredibly minor weakness - I was surprised not to see TREC mentioned in the history of field-defining benchmarks. I recognize that it's not directly related to the problems the paper is attacking head-on, but information retrieval is not _unrelated_ to natural language processing.

Finally, the conceit in the title and conclusion is fantastic and I'm in favor of keeping it. But I think you need to build the bridge for the reader from the metaphor to your arugment. That's easy - you just need to say more explicitly that benchmarks are inside the museum but the real work of the evaluated models is outside (or something to that effect).

**Additional Feedback:**

I reiterate that my goal in writing this review is not to nitpick or critique, but to offer places where the argument can be improved because it is important and deserving of a platform. But I want it to be more definitively able to occupy the space it is creating when it gets there.

**Clarity:**

Small typo-level nitpicks:
* on p1: "due to its" -> "due to their" (agreement)
* The sentence in the last paragraph of 4.1.1 is confusing - I initially read it to mean the opposite of what it says.

**Correctness:**

As noted in the "weaknesses" section, my concerns are more about completeness than correctness with two clear exceptions:
1. On p3, the paper notes how to apply construct validity to understand the core critique. Specifically, the paper says "we can think of benchmarks as mirroring the role of the _experimental setting_ in the evaluation process". But there is no experiment in ML, in the sense that there is no "hypothesis" being tested. Thus, any epistemic error is not introduced by the benchmarking process, but by the problem formulation and dataset generation processes, which have already taken place by the time a benchmark is produced or pursued. A good treatment of this issue at both the technical and philosophy-of-science level can be found in Malik, 2020 ("A Hierarchy of Limitations in Machine Learning"). The paper also says here that construct validity should be considered as an external validity issue (which is absolutely correct), but in fact there are important internal validity issues that must also be considered, such as the propensity of benchmark leaderboard chasing to lead to overfitting and poorer generalized performance (an easy technical argument here is to focus on well understood problems with boosting combined with the fact that the winning models in many leaderboard contests - e.g., the Netflix prize and many Kaggle contests - are boosting-based or were before deep learning ate the mind of the community). Malik does a good job grounding the way that standard ML workflows will lead to optimistic performance in such scenarios and relating it back to the difficult problem of bounding epistemic error. Also, a discussion of construct validity could/should engage with the broader idea of the Theory of Measurement (see, for example, Jacobs & Wallach, 2020 ("Measurement and Fairness"); Jacobs, 2021 ("Measurement as Governance of and for Responsible AI"); and a number of recent works on problem formulation in ML such as Amironesi et al., 2021, which has many overlapping authors with this paper! This is obviously an incomplete list of potential citations).

2. On p5, the paper notes the distinction between intensional and extensional specification, a longstanding and important distinction in the domain of formal program specification for the purpose of verification (bounding the specification-implementation gap which always exists for any computational process in at least some form). This is excellent, but I think the translational value could be stronger (e.g., instead of only citing a recent ML work, the authors could also identify that this is an old idea in a related field and gesture broadly in the direction of references on this, such as (I didn't look very hard here, so there might be better cites to be found) Carnap's original 1947 formulation in formal logic or the summary of how it applies in software engineering given by Eden and Kazman, 2003. Additionally, the paper goes on to state that "[i]n machine learning, the tendency is for [extensional specification]". This is absolutely incorrect, but only because it needs to be strengthened. ML is a purely extensional paradigm. Malik, 2020 does a good job explicating this more deeply, but to the extent that there is intensional specification happening in approaching an ML benchmark, it happens during the problem formulation and dataset development stages. I think this needs to be pointed out, at a minimum.

**Documentation:**

This section doesn't seem relevant to this paper, which offers a critique of benchmarking rather than a dataset or a benchmark.

**Ethics:**

When improved, I think this paper has the potential to have strong positive ethical implications for the field, and I commend the authors on the work they are doing to improve outcomes.

**Relation To Prior Work:**

In general, the work is well contextualized. I've offered a few specific citations I think would be useful in places, most notably Malik's 2020 preprint "A Hierarchy of Limitations in Machine Learning", which has a lot to speak to across my suggestions since it covers both the philosophy of science issues the paper engages with and the technical issues (e.g., the natural optimism of cross-validation-based performance measures) in extreme detail.

**Summary And Contributions:**

The paper argues that the fundamental nature of benchmarking is a limited approach to understanding general problems, even when solving these general problems are the desired outcome for the technologies undergoing benchmark evaluation. The critique offered is strong, but misses some key fundamental criticisms available and avoids taking certain positions which would make its argument stronger (see main review).

I want to love this paper. I really do. However, I worry that, as presented, it runs the risk of being ignored as irrelevant criticism by precisely the audience it should be reaching. So the authors (and other reviewers) should take anything I say which is critical as a suggestion to improve the paper. I'd really like to see it sharpened into the pointed argument I think it can be, and my score reflects my desire to see it so strengthened prior to archival publication.

Finally, I note that a version of this paper was part of the NeurIPS 2020 workshop ML-RSA (ML Retrospectives, Surveys, and Meta-analyses). That non-archival venue doesn't preclude publication in this track, but I'd be curious to see a response from the authors about what they learned workshopping that paper over the past year as a way of validating some of my concerns about reaching the target audience or missing key existing lines of criticism on this topic.

---

> ### Author Response · Authors · 2021-09-28
> **Thank you so much for such a thorough and thoughtful review! [1/2]**
>
> Thank you so much for such a thorough and thoughtful review!
>
> >  I note that a version of this paper was part of the NeurIPS 2020 workshop ML-RSA (ML Retrospectives, Surveys, and Meta-analyses). That non-archival venue doesn't preclude publication in this track, but I'd be curious to see a response from the authors about what they learned workshopping that paper over the past year as a way of validating some of my concerns about reaching the target audience or missing key existing lines of criticism on this topic.
>
> We learnt quite a bit from our early workshopping of this paper and several in-depth discussions with several peers - including a couple co-authors of GLUE, and research experts working extensively with ImageNet.
>
> What we mainly learnt from these engagements was that the widely shared claims of these benchmarks' ability to measure general capabilities often did not match author intent. Rather, such claims were disproportionately exaggerated and amplified by the ML community itself, at times to the disappointment of those involved in their construction. The current situation is thus more a convergence of several ill-formed community incentives, rather than the outcome of any particular author’s intent. Benchmark creators were thus quite open to the critiques we present in this paper.
>
> Most of the feedback we received shaped our arguments to focus on the damage being done by the rhetoric of these benchmarks claiming to address “general” performance capabilities, rather than critiquing the task of “general” performance itself. Some in the field are convinced by the goal of AGI, but many others just see the practical utility of benchmarks that can assess model generalization capabilities. Our goal in this paper is thus not to say that we never want to build systems agnostic to a particular task, scientific question or application setting - some see that as still interesting and useful work, and we didn’t want to get distracted confronting those motives. Instead, what we realized we wanted to convey much more was how such broad capabilities could not be appropriately instantiated in a data benchmark, and that this understanding should guide how we develop as well as make use of benchmarks in ML.
>
> > It buys too much into exactly the current ML discourse/hype it means to debunk.
>
> We do not take a position in the paper regarding the validity of the aspiration to general performance in ML. What we instead focus on is how, under the current dominant evaluation paradigm of benchmarking, such claims cannot be made using benchmark datasets falsely presented as being able to assess these "general" abilities.
>
> > existing benchmarks not only aren't good operationalizations of the kind of generalization performance benchmark authors and leaderboard participants say they are, but can't be for the reasons already offered.
>
> Thank you for this comment. Yes, this is a point we hoped to make in the paper, and hope to make more explicitly in a future version.
>
> > Related management and political economy issues - why do we use certain benchmarks as proxies? Who chooses this and who benefits from those choices? Does the performance construct of interest make sense at a technical/construct validity level, or is it merely a construct of convenience (as opposed to an operationalization of convenience, as the paper already discusses).
>
> The politics of benchmarking decision-making and adoption is certainly interesting, but we felt it was out of scope for this particular paper. Notably, there is some related work that begins to answer these important questions [1,2, 3,4] - we will gladly point to this work in the final version of this paper.
>
> 1. Scheuerman, Morgan Klaus, Emily Denton, and Alex Hanna. "Do Datasets Have Politics? Disciplinary Values in Computer Vision Dataset Development." arXiv preprint arXiv:2108.04308 (2021).
> 2. Dotan, Ravit, and Smitha Milli. "Value-laden disciplinary shifts in machine learning." arXiv preprint arXiv:1912.01172 (2019).
> 3. Denton, Emily, et al. "Bringing the people back in: Contesting benchmark machine learning datasets." arXiv preprint arXiv:2007.07399 (2020).
> 4. Denton, Emily, et al. “On the Genealogy of Machine Learning Datasets: A Critical History of ImageNet.” Big Data & Society, July 2021, doi:10.1177/20539517211035955.

---

> > ### Author Response · Authors · 2021-09-28
> > **Thank you so much for such a thorough and thoughtful review! [2/2]**
> >
> > >  Did not explicitly mention either the body of literature on generalization error and robustness or the body of literature on understanding epistemic error. Did not see any discussion of concept drift or other forms of practical drift which are well studied. No mention that high accuracy might belie highly unbalanced error rates for small classes or otherwise under-report performance for rare cases.
> >
> > We believe that we do mention the dangers of a single metric explicitly and point to how that disguises the presentation of performance - you make a good point to generalize this argument beyond the scope of fairness discussions and we will make sure to do so in a future version.
> >
> > Generalization and robustness error, as well as notions of practical data drift (including comments on temporal draft) were more prominently featured in a previous draft and we can bring up this discussion again briefly in our final version. Our rationale for cutting that discussion short was that we wanted to focus on the details of design and presentation for these benchmarks - namely not to necessarily discuss all the limitations of data benchmarking in general but specifically speak to the research practices that we saw as prohibitive to being able to adequately make use of benchmarks and avoid false claims to measure general capabilities. We wanted to focus our critique on the research and data engineering practices that led to the current claims and benchmark formats, so we could clearly advocate for the cultural changes necessary for our evaluation ecosystem to improve.
> >
> > >  I can't tell where the authors stand on the idea of general performance abstractly. focusing the criticism either on the goal or on the way the goal is operationalized. benchmarks do a poor job of testing knowledge about the underlying world (as the paper mentions) or because the task of "understanding" is difficult to operationalize or because it's poorly formed as a problem
> >
> > We purposefully do not take a position on the objective of general performance as a goal for ML. This objective is notoriously vague, and there are contested priorities and interpretations of this objective even amongst those that hope to achieve this goal. Instead, we wished to make a more specific and  grounded claim that the datasets presented to embody this goal do not actually do so. In fact, we believe benchmarking as an evaluation method is quite incompatible with measuring vague and high-level “general” objectives due to the inherent nature of data (as we argue in Section 4). This more precise discussion is what we hope to contribute to.
> >
> > > the relevant task is language modeling or performance on some specified task
> >
> > Modeling human language is compatible with the task of language understanding though you are correct in pointing out that both goals are not the same [1]. The goals of NLU are often articulated as the latter, though many techniques often appear to focus on the former as a primary strategy. In this paper, we adopt the understanding that NLU is meant to address the functional task of understanding rather than just the mimicry or appearance of language behavior.
> >
> > 1. Bender, Emily M., and Alexander Koller. "Climbing towards NLU: On meaning, form, and understanding in the age of data." Proceedings of the 58th Annual Meeting of the Association for Computational Linguistics. 2020.
> >
> > > The paper also says here that construct validity should be considered as an external validity issue (which is absolutely correct), but in fact there are important internal validity issues that must also be considered, such as the propensity of benchmark leaderboard chasing to lead to overfitting and poorer generalized performance.
> >
> > Construct validity is broadly thought to be an external validity issue and is at the crux of the argument we want to make here regarding how well these benchmarks could possibly represent a vague “task” involving general capabilities. We acknowledge that there are other limitations to benchmarking as a practice and many internal validity issues that plague ML evaluation more broadly but we hoped to focus on construct validity as it related to our primary argument of benchmarks as a poor operationalization of “general” objectives.
> >
> > > Build the bridge for the reader from the metaphor to your argument. That's easy - you just need to say more explicitly that benchmarks are inside the museum but the real work of the evaluated models is outside (or something to that effect). Intensional and extensional specification - instead of only citing a recent ML work, the authors could also identify that this is an old idea in a related field and gesture broadly in the direction of references. Clarity section, and named typos.
> >
> > Thank you very much for this feedback, we completely agree and will make sure to edit the text to address these considerations as well as include the necessary additional citations.

---

> > > ### Comment · Reviewer_AZHN · 2021-09-29
> > > **Do say something about generalization!**
> > >
> > > I appreciated the engagement in this response with my points very much. I still think that the paper should take a stronger position on the merits of generalization, and that this is possible while hewing to the choice (which I think is proper - arguing that general performance is an unreachable goal will bring an avoidable negative response, while arguing for more precision just improves the discussion). Specifically, there's a small bridge from your argument that I think is necessary and missing, relating back to my point about optimism in evaluation. Lab evaluation is always optimistic, since it makes the necessarily insufficient assumption that lab data represent data that will be seen during deployment. So "performance" (however measured) will decline as a system is used, first as a step function when transitioned from a training environment to a live environment and then over time due to drift. If the paper says this explicitly, I think it will connect your critique of general performance measurement to the lived experience of performance assessment more vividly.
> > >
> > > Separately, my point about rare phenomena and disparate subclass error wasn't meant to be an allusion to fairness, though it definitely is. I really did mean that ML misses rare phenomena, often _precisely because it is generalizing_. For example, a smooth decision boundary won't capture unusual cases well, but a higher-order complex decision boundary won't generalize as well (the usual bias/variance tradeoff). Here again, I think reminding the community to be explicit about assumptions and precise about what measures mean will increase the impact of your excellent work.

---

### Decision · Program_Chairs · 2021-10-09

**Decision:**

Accept

**Comment:**

The paper is very clear in its arguments against the over-reliance of machine learning on benchmarking, which it claims as used as inadequate proxies for generalized learning. This core argument is supported by historical background and suggestions for improvement, although some of these suggestions could be moved to the main body of the paper or otherwise reorganized for increased visibility.

The reviewers provided very thorough reviews. They are in agreement that this paper is well done and covers an important topic that is relevant to the track. The authors' responses to the reviews were detailed and the resulting conversation makes it clear that the paper has the potential to spark important discussions in the community.